# Anti-Bacterial Action of Plasma Multi-Jets in the Context of Chronic Wound Healing

**Thomas Maho** [1], **Raphaelle Binois** [2], **Fabienne Brulé-Morabito** [3], **Maryvonne Demasure** [2], **Claire Douat** [1], **Sébastien Dozias** [1], **Pablo Escot Bocanegra** [1], **Isabelle Goard** [2], **Laurent Hocqueloux** [2], **Claire Le Helloco** [2], **Inna Orel** [1], **Jean-Michel Pouvesle** [1], **Thierry Prazuck** [2], **Augusto Stancampiano** [1], **Clément Tocaben** [1] and **Eric Robert** [1,*]

[1] GREMI, UMR 7344, CNRS/Université d'Orléans, CEDEX 2, 45067 Orléans, France;
thomas.maho@univ-jfc.fr (T.M.); claire.douat@univ-orleans.fr (C.D.); sebastien.dozias@univ-orleans.fr (S.D.);
pablo.escot@univ-orleans.fr (P.E.B.); inna.orel@univ-orleans.fr (I.O.);
jean-michel.pouvesle@univ-orleans.fr (J.-M.P.); augusto.stancampiano@univ-orleans.fr (A.S.);
clement.tocaben@etu.univ-orleans.fr (C.T.)

[2] Centre Hospitalier Régional d'Orléans, 14 Av. de l'Hôpital, 45100 Orléans, France;
raphaelle.binois@chr-orleans.fr (R.B.); maryvonne.demasure@chr-orleans.fr (M.D.);
isabelle.goard@chr-orleans.fr (I.G.); laurent.hocqueloux@chr-orleans.fr (L.H.);
claire.lehelloco@gmail.com (C.L.H.); thierry.prazuck@chr-orleans.fr (T.P.)

[3] Centre de Biophysique Moléculaire (CBM), CNRS UPR 4301, 45071 Orléans, France; fabienne.brule@cnrs.fr

[*] Correspondence: eric.robert@univ-orleans.fr

**Featured Application: The objective of the work is to assess the use of non-thermal multi plasma jets for the treatment of infected and chronic wounds in the context of hospital centers.**

**Abstract:** This work is a contribution to the development and implementation of non-thermal plasma technology for decontamination in the perspective of nosocomial and chronic wound innovative therapies. Multi jets devices based on Plasma Gun® technology in static and scanning operation modes and bacterial lawns inoculated with resistant and non-resistant bacterial strains were designed and used. A pilot toxicity study exploring plasma treatment of wound bearing patients, performed with a low voltage plasma applicator, is documented as a first step for the translation of in vitro experiments to clinical care. Bacterial inactivation was demonstrated for *Staphylococcus aureus*, *Pseudomonas aeruginosa* and drug resistant *S. aureus*, *P. aeruginosa* and *Escherichia Coli* strains collected from patient wounds at Orleans (France) hospital. A few square centimeter large contaminated samples were inactivated following a single plasma exposure as short as one minute. Samples inoculated with a single but also a mix of three resistant pathogens were successfully inactivated not only right after their contamination but for mature lawns as well. Similar bactericidal action was demonstrated for antibiotic-resistant and non-resistant *P. aeruginosa*. The time exposure dependent increase of the inhibition spots, following multi jets exposure, is discussed as either the accumulation of reactive species or the likely combinatory action of both the reactive species and transient electric field delivery on inoculated samples.

**Keywords:** plasma jets; non-thermal plasma; plasma decontamination; resistant pathogen; wounds; plasma medicine; *Escherichia Coli*; *Staphylococcus aureus*; *Pseudomonas aeruginosa*; multidrug-resistant bacteria

## 1. Introduction

When introducing new people to "plasma" as "ionized gas", one mentions that plasma is the fourth state of matter and represents more than 99% of the baryonic matter of the universe. Natural plasmas are fascinating (aurora), a unique source of energy on earth (sun power), at the origin of basic mater physics (stars, quark plasma, amino acid generation as the bricks of life), but they are also somehow "hostile", generating huge transient

currents and overvoltage in lightning, electromagnetic disturbance in the pole region, intense energetic radiation fluxes constraining microorganisms, and life expansion. This was indeed one the first historical developments of plasma science and technology to try to benefit from the radiative (UV, X-rays) and chemical compounds (oxidants, metastable states, charged particles with high energy) of plasmas for decontamination or sterilization purposes. Besides the pioneer works [1–3] and industrial device development [4,5] based on plasma generation, innovative ongoing research works are still continuously reported on this topic [6–9]. New eco-friendly alternatives pushed by consumer needs and environmental challenges, such as the use of natural ingredients for cosmetics, the design of innovative food product packaging, food product shelf life lengthening, and also the emergence of antibiotic-resistant pathogens or new viruses are key examples where an increasing need for new decontamination strategies are continuously required. Many among them are based on plasma technology [10–13]. Indeed, plasma are known to be a potential and likely unique strategy for resistant microorganism reduction, mostly due to plasma's multifaceted features, merging energetic radiation, energetic particles, and chemically active compound cocktails likely combining short and long-lived oxidants.

In this context, non-thermal plasma, or so-called cold plasma, has been developed in the past decade for plasma biomedical applications. Among them, the therapeutic applications in "plasma medicine" [14,15] emerge, the two probably more advanced fields being wound care [16–23] and new approaches for cancer treatment [24–31].

Clinical studies dedicated to wound healing indicate that pathogen abatement, tissue growth stimulation and immune response are three of the modes of action that plasma delivery can trigger and control. Pathogen reduction is a key issue in hospitals where more and more multidrug-resistant strains of bacteria are emerging, thus limiting or even preventing the use of conventional disinfectants or drugs. Brun et al. [32] reported on plasma's antibacterial effect with an RF helium afterglow on multi resistant *Pseudomonas aeruginosa* and methicillin-resistant *Staphylococcus aureus* exposed in vitro samples. Planktonic or biofilm growth on borosilicate slides cultured in 24-well plates were considered in this work, and a combination of plasma treatment with antibiotics was also evaluated. In [33], a dielectric barrier discharge (DBD) plasma torch generated with an air flow at 3.5 L/min and delivering a plasma jet 4 mm in diameter was used to study the sensitivity of antibiotic-resistant and non-resistant bacteria. The microorganisms were set in a thin liquid layer covering a glass substrate. The authors conclude that plasma degrades components (DNA) of the bacteria so that the plasma method may be useful in eliminating bacteria that are resistant to conventional antibiotic therapy. The application of DBD plasma was also demonstrated as a valuable decontamination technique for the removal from inert surfaces of planktonic and biofilm-embedded bacteria such as methicillin-sensitive *S. aureus* (MRSA-USA 300, MRSA-USA 400) and *E. coli*, the most common hospital contaminants [34]. A plasma source named FlatPlaSter2.0., consisting of surface micro-discharges generated in air over a surface of $13 \times 9$ cm$^2$ was assessed as an alternative disinfectant for dry surfaces [35]. Bacterial endospores including *Clostridium difficile* and vegetative bacteria including *Enterococcus faecium* were considered. It was concluded that air plasma can disinfect dry inanimate surfaces contaminated with bacteria, including bacterial endospores. The effect was mainly limited by the bacterial density. The perspective demonstrated that plasma could serve as an alternative in the disinfection of medical instruments complementary to the standard cleaning procedures. He et al. [36] reported on the sensitivity of two drug-resistant bacteria to low-temperature air plasma in catheter-associated urinary tract infections. Low temperature air plasma demonstrated efficiency to decontaminate resistant *E. coli* and *Enterococci* through the destruction of the ribosome and other organelles inside the bacteria. Besides all of these plasma and plasma afterglow treatments, two other protocols can be envisioned for decontamination applications. They consist of plasma treated solution delivery or aerosol injection in combination with plasma discharge [37,38]. Indeed, in the context of the COVID pandemic and with the perspective of future applications in controlling the microbiota in tubes and tracheal appliances used in the respiratory tract,

the anti-microbial efficacy of nebulized plasma activated water applied for 15 min shows the inactivation of *S. aureus* and *E. coli* [39].

Publications dealing with resistant micro-organisms in the context of clinical protocol are much more scarce. Nevertheless, Daeschlein et al. [40] demonstrated that argon-based cold atmospheric plasma can serve as a potent treatment modality to reduce and even eradicate numbers of multidrug resistant pathogens colonizing patient' wounds. This outstanding work was performed with a "plasma cold beam" fed at 6 L/min with argon and resulting in a target plasma spot of about 5 mm in diameter. Heinlin et al. [41] reviewed plasma applications in medicine with a special focus on dermatology. Among various topics, they report 291 treatments in 36 patients, and found a highly significant reduction in bacterial load in wounds treated with argon plasma compared with untreated wounds. This reduction was found in all types of bacteria, even in multidrug-resistant bacteria such as methicillin-resistant Staphylococcus aureus (MRSA). Following these encouraging reports, there now exists a great interest in developing dedicated non-thermal plasma sources that are efficient and easy to implement in hospital or medical centers to face the critical challenges of nosocomial infection and chronic wounds.

In this work, preliminary results are documented on the use of non-thermal plasma jets for bacterial load reduction in the context of chronic wound therapy. The following section presents the various plasma multi jet devices developed for this work and the bacterial targets used for the evaluation of the plasma decontamination opportunities. Afterward, in vitro decontamination assessments are presented and discussed for various pathogens, including resistant strains collected from patients at the Orleans hospital (CHRO). Finally, preliminary results obtained with low voltage plasma jets in the context of the toxicity evaluation for patients' treatment through a pilot study undergone at Orleans Regional Hospital in the department of Tropical and Infectious Diseases are briefly introduced.

## 2. Materials and Methods

### 2.1. Plasma Jet Devices

This work is based on the use of non-thermal plasma devices for in vitro applications and a pilot study. Based on our know-how at GREMI, the plasma gun technology [42,43] that was used in this work needed to meet two main requirements: to develop plasma jets while keeping the temperature of the target sample below 40 °C, and to afford the best strategy to expose large contaminated surfaces during short time treatments. With regard to the treatment of infected and chronic wounds, the typical surfaces to be exposed are of a few to tens of square centimeters large and exhibit structured three dimensional topologies, i.e., they do not necessarily consist of flat targets. Thus, the whole surface exposure should be performed with likely significant variations of the distance between the plasma jet and the sample and the full time treatment should not exceed a few minutes.

The first option, recently reported in [44], is to expose large surfaces to a room temperature single plasma jet powered with high frequency (typically 10 to 20 kHz with our device) voltage pulses of microseconds in duration. The tuning of the voltage peak amplitudes, gas flow rates, and the consideration of the target electrical conductivity features was demonstrated to allow for the controlled exposure of a surface with a typical diameter of 2 cm. Nevertheless, the larger surface exposure was not easy to achieve without an additional displacement of the plasma plume, and this device was to be operated at voltages ranging from 10 to 20 kV.

The second option consisted in the operation of so-called multi-jets generated from a single primary jet [43,45,46]. These multi-jets were used for the in vitro study performed in this work with two applicators: the one delivering multi-jets in ambient air leading to a multi-spot impact, and the other using the same device but with an additional sleeve leading to helium accumulation over the target, while not being fully airtight to maintain reactive oxygen and nitrogen species generation. This second applicator results in plasma delivery as a combination of multi-spots and a diffuse plasma generated in between the

individual jets. These two multi-jet devices are called "multi-spot" and "diffuse mode" applicators in the following analysis. Figure 1 presents the schematics of these two devices.

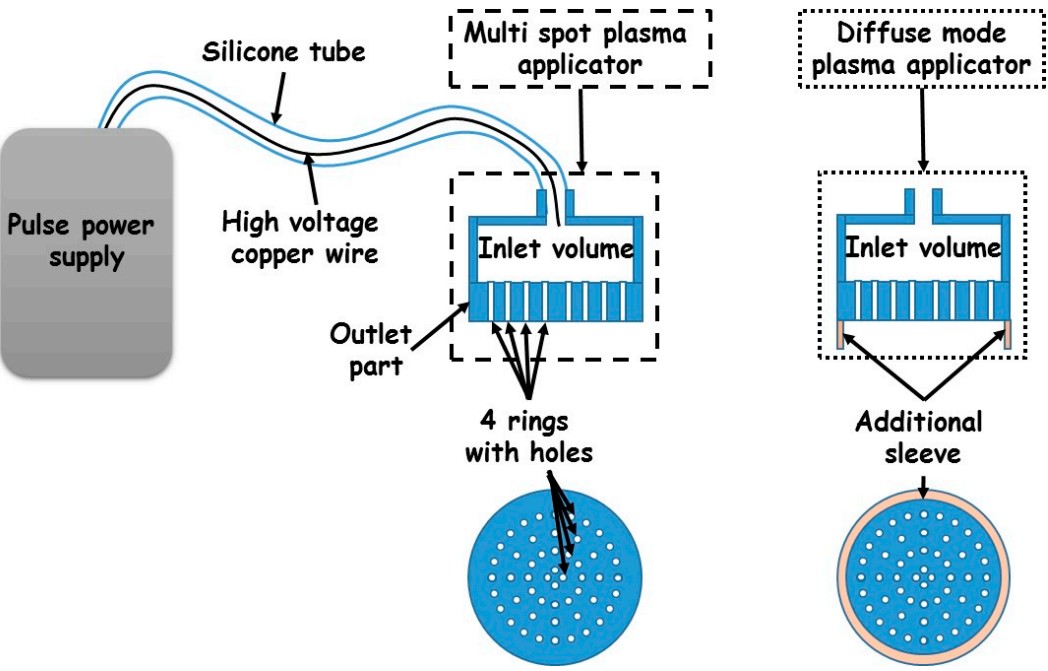

**Figure 1.** Schematic of the multi-jets devices (option 2) with pulse power supply, high voltage wire inside the silicone and center: multi-spot, and right: diffuse mode applicators. On bottom end on view of the applicators. When operated, the silicone tube connection for the diffuse mode applicator is the same as for the multi-spot applicator.

As shown in Figure 1, any of the two plasma applicators was connected through a flexible silicone tube inside which high purity (99.999%) helium was flushed at 4 L/min. The pulse power supply generates voltage pulses of 4 μs FWHM, having a peak amplitude of +14 kV and delivered at a 2 kHz repetition rate. These voltage pulses were delivered via an isolated copper wire inserted all along the silicone tube to the tip located at the inlet of the shower-like multi-jets applicator schematically documented in Figure 1. This remote plasma jet generation, including a flushed gas and high voltage wire inside a flexible tube, was previously reported for endoscopic application [47] and multi-jets generation [48]. The shower-like applicator consists of a dielectric (Delrin ®) assembly having an inlet volume (50 mm in diameter, 20 mm height) and an outlet part equipped with 52 channels (0.8 mm in diameter, 10 mm height). These 52 channels are distributed along four rings of 8, 18, 28 and 38 mm in diameter and having 4, 12, 12 and 24 holes, respectively. The discharge is ignited at the tip of the high voltage wire as a primary helium plasma jet. This primary plasma jet propagates in the inlet volume of the shower-like applicator before splitting into secondary helium jets generated in the channels of the outlet section of the applicators [46]. With the multi-spot plasma applicator, all of the secondary jets distributed along the 8, 18, 28 and 38 mm diameter rings were delivered simultaneously. The plasma that was in contact with the sample is associated with ionization wave propagation in the helium gas flow at the outlet of the capillary.

The third option consists in the generation of multi-jets in a bunch of individual helium flushed capillaries, each equipped with an inner needle-like powered electrode, and a grounded ring electrode surrounding all the 4 mm inner diameter, 3 cm long dielectric capillaries. The short distances between the needle tip and the capillary outlets (5 mm in this work) and the short gap from the capillary outlet to the samples (a few mm in this work) allow the generation of the multi-jets with voltage pulses of lower peak amplitudes of 2 kV. This low voltage operation enables the use of a more compact and lower power-consuming pulsed power supply and operation at a higher pulse repetition rate (20 kHz

in this work) while keeping the input power to the multi-jets applicator in the range of a few watts. Another key advantage of this third class of multi-jets, as was shown, is that the distance to the samples is much less critical for each individual jet's generation over non-flat targets, which is representative of chronic wounds. In a first step, this type of device has been used for the pilot study in a single jet configuration as will be described in the results section. The multi-jets configuration for the generation of four secondary jets is shown in Figure 2. This third class of multi-jet device is called the "low voltage" plasma applicator, as in the following.

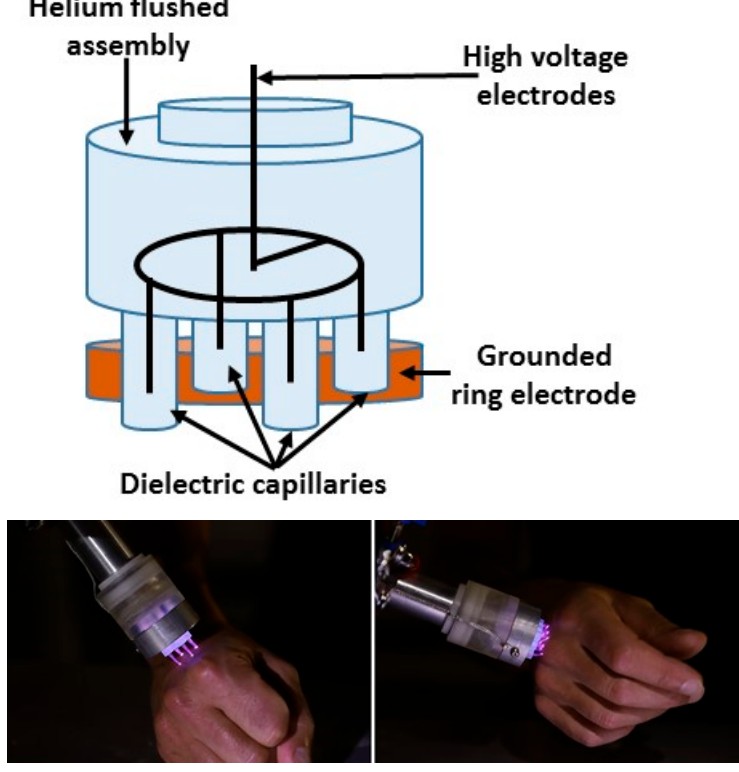

**Figure 2.** Top: schematic of the low voltage multi-jet applicator (option 3). Bottom: 4 and 9 low voltage multi-jets delivered on the author's hand. Parameters: voltage 2.5 kV, repetition rate 20 kHz, gas flow 2 L/min.

The Table 1 summarizes the main characteristics for the three plasma device options discussed in this section.

**Table 1.** Main characteristics of the plasma gun based devices.

|  | 1st Option Single Jet | 2nd Option Multi-Jets from One Primary Jet | | 3rd Option Bunch of Individual Jets |
|---|---|---|---|---|
| Voltage (kV) | 10–20 | 14 | | 3 |
| Repetition rate (kHz) | 20 | 2 | | 10–20 |
| Gas flow (L/min) | 0.5 | 4 | | 3 |
| Key advantage | Simplest design | Large surface treatment | | Less sensitive to target distance topology |
| Name of the plasma applicator | Single jet | Multi spot | Diffuse mode | Low voltage |

**Table 1.** *Cont.*

| | 1st Option Single Jet | 2nd Option Multi-Jets from One Primary Jet | | 3rd Option Bunch of Individual Jets |
| --- | --- | --- | --- | --- |
| Plasma impact | One two mm spot moving over a 2 cm² disk | 4 + 12 + 12 + 24, one mm spots distributed over 8/18/28/38 mm in diameter rings | Same as multi spot with additional diffuse volume between the multi jets | 4/9/13 two mm spots distributed over a 2/3/3 in diameter disk |

*2.2. Biological Samples*

2.2.1. Bacterial Lawn Preparation and Plasma Treatment

As shown in Figure 3, a membrane filtration system composed of a funnel and a vacuum valve was used to prepare samples with different bacterial strains. The bacterial inoculum solution was set in the funnel, the base of which was equipped with a membrane 47 mm in diameter and having a porosity of 0.45 μm. This membrane, set on a filter stone, acted as filter for the inoculum when the vacuum pump was switched on. In this work, an inoculum ($10^3$ CFU/mL) volume of 100 mL was filtered across the membrane. The membrane was then set on an agar disk (50 mm in diameter) prepared in a petri dish. Following a 24-h incubation period at 37 °C, a 10 cm² bacterial lawn was produced as shown in Figure 3d.

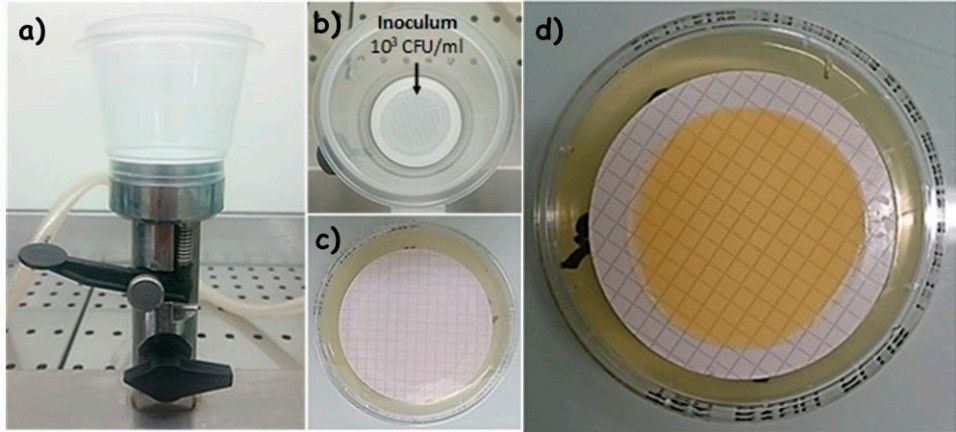

**Figure 3.** (**a**) Funnel and valve connected to a vacuum pump (not shown) used to initiate the bacteria inoculum preparation, (**b**) Top view of funnel, whose base is covered with a membrane used to filter the bacterial inoculum, (**c**) zoom on the membrane set on agar prepared in a petri dish right at the end of the filtering process, (**d**) view of the grown bacterial lawn following a 24-h incubation at 37 °C.

As mentioned in the results section, the plasma treatments were performed right after the inoculated membrane was set on the agar substrate, i.e., before the 24-h incubation period, meaning before the growth of the bacterial lawn.

The plasma treatments consisted of 1, 5 or 10-min exposure with a gap distance between multi-jet bottom base and the membrane of 5 mm, a gas flow rate of 4 L/min, peak voltage amplitude of 14 kV and pulse repetition rate of 2 kHz (both multi-spot and diffuse mode applicators were used). All experiments have been performed in triplicate. Control samples consisted in gas flow only exposure of the membrane on agar substrates for the same durations (1 to 10 min) but with no plasma ignition.

A larger surface bacterial lawn of 12 × 12 cm² inoculated with patients' collected resistant bacteria have also been prepared and exposed to plasma. A 100 μL volume of bacterial solution having $10^4$ colony forming units per milliliter (CFU/mL) was spread over the surface of the agar plate. These large surface inoculated agar samples were then incubated at 37 °C for 24 h with or without preliminary plasma treatment. The plasma treatment with the multi-spot applicator was performed in a scanning protocol with the

help of a programmable displacement board allowing various scanning patterns and speeds with the fixed plasma applicator and the moving agar sample as shown in Figure 4.

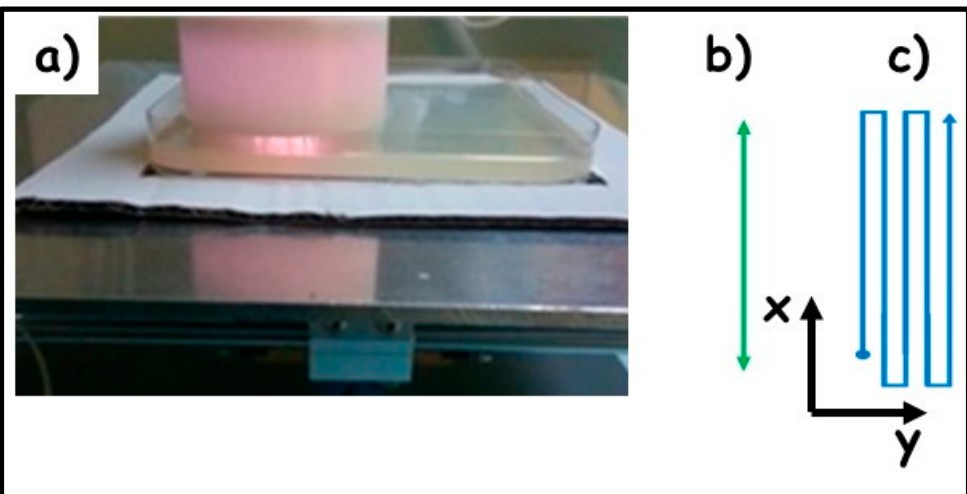

**Figure 4.** (**a**) Multi jets treatment of large surface inoculated agar samples set on a programmable displacement board, (**b**) translation scanning pattern, (**c**) zigzag scanning pattern.

Two scanning patterns have been experienced. First was a translation pattern with back and forth linear displacements, the back and forth duration being 12 s and having an extension of 100 mm. This translation pattern was repeated respectively 5, 25 or 50 times, thus corresponding to a whole plasma treatment duration of 1, 5 and 10 min. The second pattern, zigzag-like, is shown in Figure 4c, and combined translation along the x and y axis of 100 and 2 mm, respectively. The time set to process the full pattern was one-minute long and this was repeated for 5 or 10 times, corresponding to 5 or 10-min plasma exposure durations.

### 2.2.2. Bacterial Strains

Two classes of bacterial strains have been used in this work. First, the freeze-dried strains, *Staphylococcus aureus subsp. aureus* ATCC®9144TM and *Pseudomonas aeruginosa* ATCC®9027TM were respectively cultured on ATCC® Medium 18: Trypticase Soy Agar (TSA) and ATCC® Medium 3: Nutrient agar (supplied by Institut Pasteur). These two non-resistant bacteria were inoculated on agar substrates from solutions having a $10^3$ CFU/mL concentration. Second, three resistant strains were isolated from wound samples collected from patients suffering from chronic wounds, namely *S. aureus*, *P. aeruginosa* and *E. coli*. The individual strain selection was achieved by culturing these patient's samples on various selective agars. These resistant bacteria were then inoculated on TSA substrates from solutions having a $10^4$ CFU/mL concentration. The experiment with the combination of these three resistant bacteria, documented in Section 3.4, was performed from mixing the three bacterial solutions prepared with a $10^4$ CFU/mL concentration of each bacteria. The resistance of each resistant bacteria strain was qualified through antibiogram analysis with the inclusion of nine different antibiotic drugs. The most resistant strain was measured to be the *P. aeruginosa* being only partly sensitive to one aminoglycoside drug. This strain was selected to illustrate the potent bactericidal action of multi jets on one resistant pathogen, as documented in Section 3.3.

All the decontamination experiments were performed in the microbiology hygiene laboratory of the Orleans Regional Hospital under the supervision of Dr M. Demasure.

### 2.2.3. Agar Samples for Multi-Jets Characterization

Agarose gel samples were prepared by dissolving 1.5 mg ml$^{-1}$ agarose of microbiology powder in a physiological saline solution (NaCl 150 mM). The solution was heated on a

conventional heating magnetic stirrer until all the powder was dissolved. Agarose mix was poured to obtain gels of 2 mm thickness. The gels were stored at 4 °C for 12 h before use. KI-starch-loaded agarose samples were used to evaluate the spatial distribution of generated reactive species. The samples were prepared by dissolving 1.5% agarose of microbiology powder, 0.3% KI and 0.5% starch $(C_6H_{10}O_5)_n$ in a saline physiological solution (NaCl 150 mM).

### 2.3. Current Measurement

Current measurements for multi-jets impinging on a grounded metallic plate documented in Section 3.1 were performed by measuring the voltage drop across a carbon resistor connected in series with the plate. The signals were recorded using a Tektronix 500 MHz probe (TPP0500B) 500 MHz bandwidth and a 500 MHz bandwidth digital oscilloscope Tektronix MDO3054.

## 3. Results

### 3.1. Multi-Jets Characterization

The power coupled to, and the current flowing across each of the individual 52 secondary jets with the multi-spot applicator were not measured in this work, but it is speculated that they are more or less evenly distributed in between each of these secondary jets. Indeed, while not specifically assessed in this work, the longtime exposure up to 5 to 10 min of the agar sample never revealed any local melting, ablation or deformation of the sample at the position of the plasma spots or in the zone in between them. This is an indication that neither severe temperature increase nor dramatic ion bombardment occur during multi-jets exposure. Nevertheless, Figure 5 reveals that the secondary jets generated from the two inner rings of the multi-spot plasma applicator exhibit a higher visible light intensity. A primary reason for this lies in the secondary jets mechanisms. Indeed, the secondary jets are generated after the primary jet ignited in the upper section of the applicator splits in the first channels distributed along the first inner ring [45,46]. Then a second splitting of the remaining ionization wave propagating in the upper section from the first to the second ring occurs and generates the secondary jets emerging from the second ring channels. The same process repeats for the third and fourth series of multi-jets emerging from the periphery of the applicator. During these splitting processes, part of the power is carried by the secondary jets, the rest being transported in the plasma developing in the upper part of the applicator before the next splits occur in the large diameter ring. A second likely reason for this higher intensity of the inner secondary jets is that even if the upper section was empirically designed to allow for a homogenous gas flow distribution in the channels where the secondary jets are generated, it is suspected that this gas flow is lower in the peripheral holes than in the center holes.

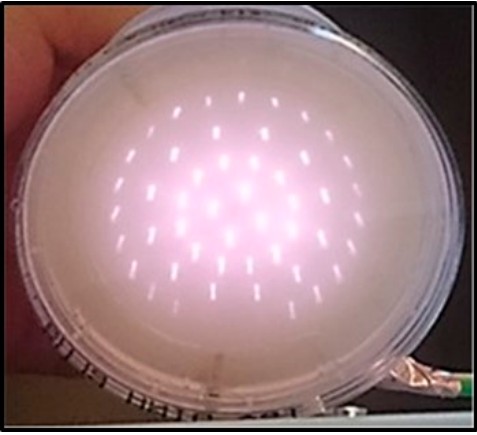

**Figure 5.** Photograph of the multi-spot plasma applicator delivering the secondary multi-jets in ambient air.

Figure 6a presents the current measurements performed with the use of a grounded metallic target set 10 mm away from the low voltage applicator delivering 4 secondary jets. By using a dielectric foil set to impede either none, one, two, and three of the secondary jets to impact the target, Figure 6b demonstrates that each one the secondary jets carry on a fourth of the whole current measured when the four jets impinge on the target.

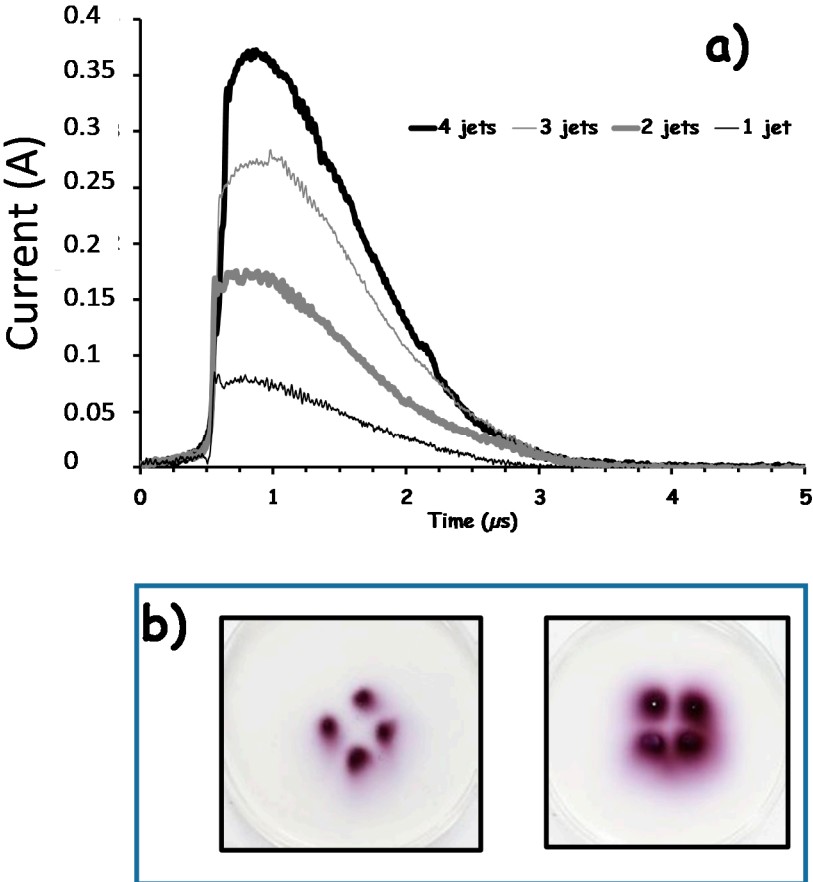

**Figure 6.** (**a**) Current delivered on a metallic target with 4 jets: bold black trace, 3 jets thin grey trace, 2 jets: bold grey trace and 1 jet: thin black trace, (**b**) image of KI-enriched agar samples exposed to four multi-jets generated with the low voltage applicator during left: 60 s and right: 240 s.

Figure 6b illustrates the operation of this four multi-jet device exposing the agar sample [49] enriched with potassium iodide. The potassium iodide is used as an indicator for the delivery of reactive species by the multi-jets [44,50]. Is it shown that for a 60 s plasma exposure, the reactive species are delivered as four spots on the sample while for a longer exposure of 240 s, the reactive species are detected almost all over a disk defined by the edges of the four multi-jet assembly. The same results were achieved for applicators generating 9 and 13 secondary jets distributed over a disk 50 mm in diameter.

### 3.2. Inactivation of Staphylococcus aureus

Figure 7 presents the multi-spot plasma treatment of *S. aureus* inoculated membrane set on agar substrates, in various experimental conditions.

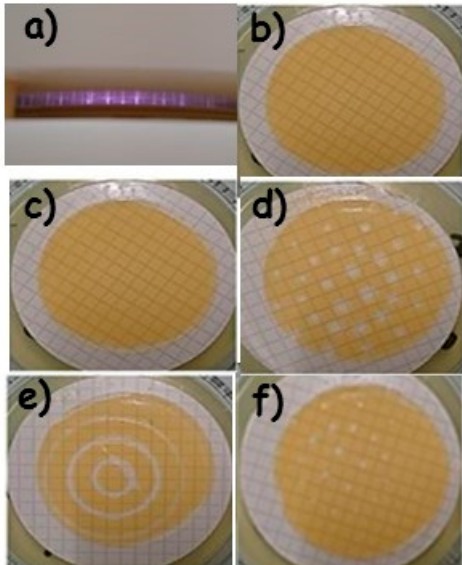

**Figure 7.** Inactivation of *S. aureus* with the multi-spot plasma applicator. (**a**) Multi-spot plasma treatment of inoculated membrane on agar substrate, (**b**) H24 image of the untreated sample, (**c**) H24 image of the sample exposed at H0 to 5 min helium flow, (**d**) H24 image of the sample exposed at H0 to 5 min plasma treatment in static mode, (**e**) H24 image of the sample exposed at H0 to 5 min plasma treatment with a 1 rpm rotation, (**f**) H24 image of the sample exposed at H0 to 1-min plasma treatment in static mode. H0 and H24 stand for hour zero as the time origin and 24 h later, respectively.

As expected, the control gas treatment shows no bactericidal action, as shown in Figure 7c where the bacterial lawn full development is observed in the same manner as for the untreated sample in Figure 7b. This confirms that the helium gas flow has no impact on the inoculated membrane, inducing no drying or spray action, as the 24-h incubation results in the same growth of the bacterial lawn as for the untreated sample. In Figure 7d, it is observed that the impact of the y 4, 12 and 12 secondary jets, distributed from the center to the periphery of the plasma applicator, results in a full inactivation of the bacteria. Because of an imperfect centering of the plasma applicator with respect to the sample, only four secondary jets distributed along the 38 mm diameter outer ring of the multi-spot applicator impacted the bacterial lawn, which was about 40 mm in diameter. This is visible in the top right region of Figure 7d. Indeed, 24 h after the multi jet treatment, the bacterial lawn growth is fully inhibited at the positions of these 32 plasma jet's impact. This first indicates that the plasma action is localized in the region where the plasma jets impact and does not induce any strong effect in between the secondary jet's regions during plasma jet exposure, but after the following 24-h incubation as well. The ratio of the sum of these 4, 12, 12 and 4 plasma spot impact surfaces over the bacterial lawn surface results in about an 8% coverage for a 5-min-long static exposure with the multi-spot applicator. Thus, the decontamination of the full surface for the extended bacterial lawn would require a displacement of the plasma applicator, as will be documented in Section 3.4. One observes that the higher intensity of the central multi-jets, as discussed in Figure 1, results in larger inactivation spots as compared to the peripheral multi-jets effect. As the visible diameters of all the secondary jets is approximately constant, this means that the plasma light emission cross section observed on the sample surface is not the best indicator to predict the inhibition zone diameter. Both plasma generated reactive species and transient electric fields inherent with the multi-jets delivery may have an effect away from the plasma emitting light volume.

Figure 7e indicates that 5 min treatment with the sample rotation (rotation speed was of 1 revolution/min) is enough to prevent bacterial lawn development at the plasma jet impact positions. This means that plasma exposures shorter than 5 min are sufficient to inactivate *S. aureus*. One also notes that the inactivation is more efficient along the two inner

circles where 4 and 12 multi-jets, respectively, are distributed, than for the third and fourth larger diameters circles exposed with the 12 and 24 multi-jets, respectively, ignited out of the channels from the two outer rings of the applicator. Two main reasons may be invoked to explain this observation: first, as discussed in Figure 5, the multi jets on the center are more intense; and, secondly, for a constant rotation speed of the sample, the plasma exposure duration per surface unit is obviously shorter for the larger diameter circles. The key role of the treatment time on the inhibition zone size is confirmed in Figure 7f, where a one-minute multi-jet exposure was processed in a static mode, i.e., without a sample rotation. This short exposure is enough to inactivate the bacteria, but the inhibition spots are much smaller and correlate with the 5 min and rotating sample observations in Figure 6e along the two larger diameter inhibition circles.

The use of the multi-spot plasma exposure protocol including the rotation of the sample, demonstrates that the increase of the number of secondary jets for larger diameters is convenient to expand the plasma applicator action on large surface and that scanning of the sample at moderate speed is efficient as will be confirmed in Section 3.4. The optimal protocol results from the balance between the goal to set a high scanning speed to allow for multi pass over the sample during a short plasma treatment duration and the limitation imposed by the necessity to prevent too strong helium gas flow disturbance at the outlets of the channels of the applicator. This latter point depends on the helium gas flowrate and the design (length and diameters) of the channels.

Another strategy assessed in this work, was to generate multi-jets in a more helium-rich volume to try to achieve a more diffuse and larger volume plasma discharge at the surface of the sample. The Figure 8a presents the diffuse mode plasma treatment of *S. aureus* inoculated membrane set on agar substrates, in various experimental conditions.

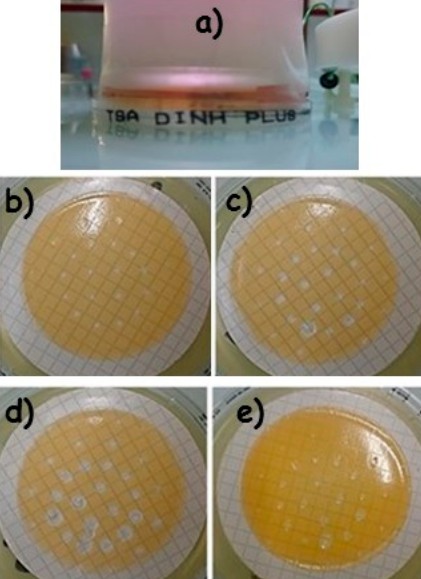

**Figure 8.** Inactivation of *S. aureus* with the diffuse mode plasma applicator. (**a**) Diffuse mode plasma treatment of inoculated membrane on agar substrate, H24 images of the sample exposed at H0 to 1 (**b**), 5 (**c**) and 10 (**d**) minute plasma treatment, (**e**) H24+10 min image of the sample exposed at H24 to 10 min plasma treatment. H0 and H24 stand hour zero as the time origin and 24 h later, respectively.

Figure 8b–d present the bacteria inactivation for 1, 5 and 10-min exposures in static mode with the diffuse mode applicator. Three main observations can be done.

First, the inhibition diameter is time exposure-dependent, indicating that the inactivation is not only the result of the direct impact of the plasma on the sample but that the plasma-induced effect includes either some diffusion mechanism or bystander processes previously speculated for plasma cancer treatment experiment analysis [51,52]. In this work, the results discussed from Figure 6 showing that the reactive species delivery in agar

sample was plasma exposure duration dependent tend to support that the inhibition zone diameter increase is likely due to the diffusion of the reactive species. Nevertheless, as during plasma jet exposure, the bacteria are also submitted to intense transient electric field during each voltage pulse, it may be suspected that the accumulation of these electric field stimulation in combination with the delivery of reactive species may play a role in the gradual spread on the inhibition zone. Indeed, the electric field driven by the space charge at the tip of the ionization wave front, such as developing for kHz plasma jets, combines a longitudinal and a radial component, the latter exhibiting a decaying intensity versus the distance from the plasma jet axis [45,53–57]. With our experimental set-up, and with a slightly higher peak voltage amplitude applied to the plasma reactor 16 kV instead of 14 kV in this work, electric field with amplitudes ranging from 3 to 8 kV/cm were reported along the plasma jet inside the capillary [45], while amplitude as high as 45 kV/cm were measured at the tip of the plasma plume [55]. One may suspect that besides the reactive species delivery and action on bacteria, the bacteria on the central zone of the plasma jet impact zone will rapidly be inactivated as there the electric field strength is large enough, while those on the more outer locations will require the accumulation of a larger number of electric field pulses of lower strength. Such combinatory and threshold-dependent action of pulsed electric field with reactive species was recently reported [48,58–61] during plasma treatment of various cell lines.

Second, the inhibition diameters are no longer larger for the four central multi-jets but for the 12 s multi-jets distributed on the 18 mm in diameter ring, conversely to the results obtained with the multi-spot applicator in Figure 7.

Third, the yellow color, representative for the *S. aureus* colonization, is less intense in between the multi-jet impact regions in Figure 8 than it was in Figure 7. This indicates that the growth of bacterial lawn even in the region in between the multi-spots is reduced with the diffuse mode applicator, presumably because of the reactive species' diffusion from impact plasma points thanks to the additional sleeve.

These two latter observations reveal that the diffuse mode applicator modifies the relative bactericidal efficiency of the individual multi-jets and that the whole surface of the sample is plasma-treated and partly inactivated even though the multi-jets are much more effective than the diffuse plasma.

In the context of wound treatment, the delivery of plasma can be planned for two different scenarios. The first would be to prevent either bacteria colonization right after the wound occurrence or bacteria re-colonization following the disinfecting care of a persisting wound. This was the motivation of the previous documented results in Figures 7 and 8, where the plasma delivery was performed right after the bacteria inoculation on the agar sample. The second and most critical need for alternative treatment, likely plasma-based, is for infected or chronic wound situations where the bacterial colonization is already achieved. To assess the potential action of plasma multi-jets for this second scenario, Figure 8e presents the 10-min-long treatment performed on a mature bacterial lawn, obtained after a 24 h incubation of the inoculated membrane on the agar substrate. It is observed that the diffuse mode applicator induces some bactericidal effect on the mature bacterial lawn. Partly inhibited multi-spots are observed together with a slightly less yellow colored lawn in the center than in the peripheral regions of the sample.

*3.3. Inactivation of Pseudomonas aeruginosa ATCC®9027TM and Resistant Pseudomonas aeruginosa Cultured from Patient's Sampling*

Figure 9 presents treatment time dependence with the diffuse mode plasma applicator for the *P. aeruginosa* ATCC®9027TM and resistant *P. aeruginosa* inoculated membrane set on agar substrates.

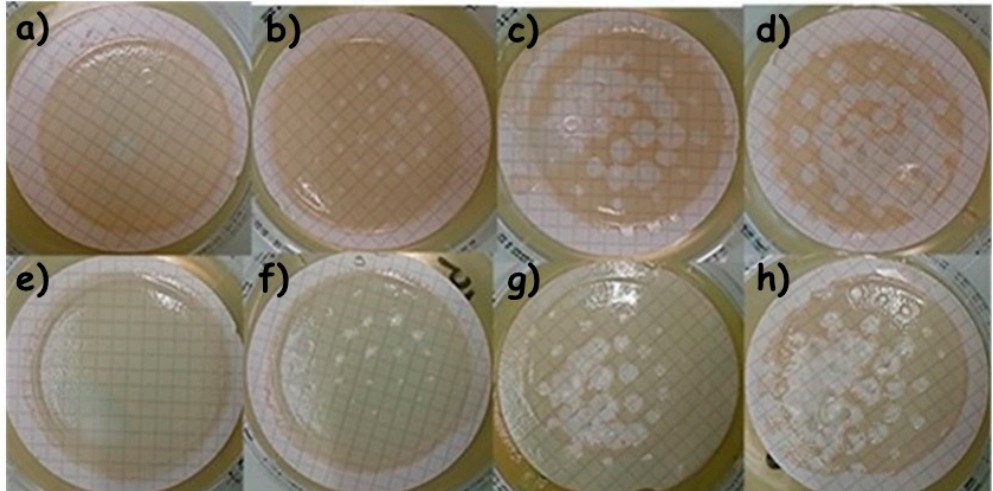

**Figure 9.** Inactivation of *P. aeruginosa* ATCC®9027TM and of resistant *P. aeruginosa* with the diffuse mode plasma applicator. H24 images of the *P. aeruginosa* ATCC®9027TM inoculated sample exposed at H0 to 0 (**a**), 1 (**b**), 5 (**c**) and 10 (**d**) minute plasma treatment and of resistant *P. aeruginosa* inoculated sample exposed at H0 to 0 (**e**), 1 (**f**), 5 (**g**) and 10 (**h**) minute plasma treatment.

As obtained with the *S. aureus* in Figure 8, the use of the diffuse mode applicator for 1, 5 and 10 min results in the inactivation of the two *P. aeruginosa* strains. It is again observed that the inactivation spot diameter, associated with each single jet, increases with the treatment duration, and that the bacteria lawn growth is also reduced in the region in between these spots, where the diffuse plasma is active. Two main additional observations can be made. First, it is quantitatively shown that the multi-jets-based treatment is more efficient for the two *P. aeruginosa* strains than for *S. aureus*. Second, it is noticeable that the multi-jet-based treatment is as efficient for the regular as it is for the resistant strain, clearly differencing the plasma action from other conventional strategies based on antibiotics supply. This was the main objective of this work to assess a new bactericidal agent in the perspectives of the development of innovative disinfecting process for situation were no therapeutic protocol is today available, as unfortunately more and more encountered with nosocomial infections.

### 3.4. Large Surface Resistant Bacterial Lawn Inactivation with Scanning Protocols

The Figure 10 presents the multi-spot plasma treatment of resistant *P. aeruginosa*, *S. aureus* and *E. coli* inoculated agar substrate, the three strains being collected from patients suffering from chronic wounds. The translation scanning mode was processed for 1, 5 and 10 min in different regions of the agar plate.

There may be some overlap of the treatment in one zone (for 1, 5 and 10 minute' plasma exposure) with the neighboring zones. Because of this possible artifact, the decontamination action is discussed for the central band of each of the three different treatment duration zones. For each of the three strains, the multi-spot scanning protocol results in a time-dependent inactivation efficiency. Nevertheless, even for the shortest, one minute treatment time, the bacterial lawn growth is almost already fully and homogenously inhibited. It is observed that the *P. aeruginosa*, *S. aureus* present a higher sensitivity than *E. coli* to the plasma treatment. Figure 10 presents the images of the bacterial lawn after the plasma treatment and a 24-h incubation period, but it was also verified that even after an incubation period as long as seven days the bacterial lawn growth was still completely inhibited.

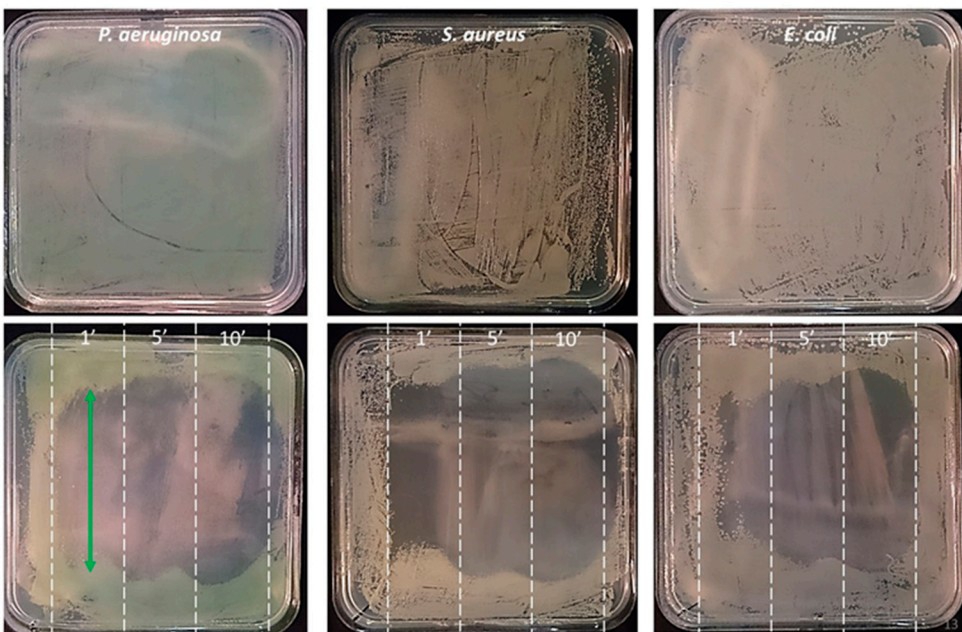

**Figure 10.** Inactivation of large surface resistant bacterial lawn with the multi-spot plasma applicator with the translation scanning mode. On top, images of resistant *P. aeruginosa* (**left**), *S. aureus* (**middle**) and *E. coli* (**right**) 24 h after their inoculation. In the bottom, H24 images of the *P. aeruginosa* (**left**), *S. aureus* (**middle**) and *E. coli* (**right**) inoculated sample exposed at H0 to 1, 5 and 10-min plasma treatment as indicated between the vertical dashed lines. The size of each treated zone is about 40 cm$^2$.

In terms of a patient's wound plasma treatment, it is important to assess the bactericidal action of plasma jets not only on a single strain inoculum but also for a mixture of different strains, as wounds are most of the time colonized by a symbiotically living bacteria assembly. Figure 11 presents the multi-spot plasma treatment of a combination of resistant *P. aeruginosa*, *S. aureus* and *E. coli* inoculum spread over an agar substrate. The zigzag scanning mode was processed for 1, 5 and 10 min in one selected region of the agar plate. This scanning mode was selected to mimic the way medical doctors or nurses may treat a large wound with a plasma device.

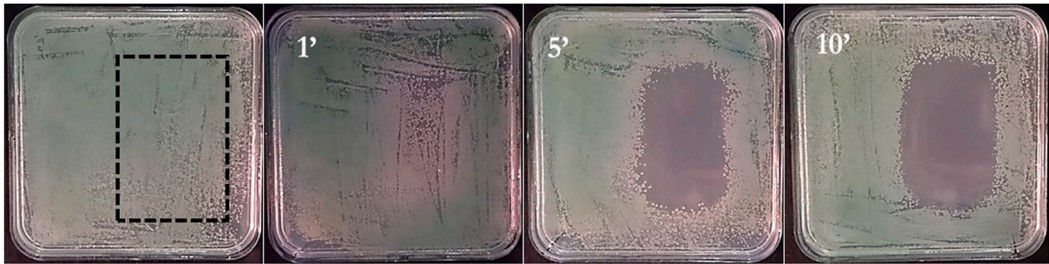

**Figure 11.** Inactivation of large surface bacterial lawn composed of a mixture of three resistant strains with the multi-spot plasma applicator applied in the zigzag scanning mode. H24 images of the inoculated sample exposed at H0 to 1, 5 and 10-min plasma treatment. The dashed rectangle zone indicates the plasma treated region. The size of the treated zone is about 32 cm$^2$.

It is observed that after a one-minute plasma treatment, the *P. aeruginosa* inactivation is already achieved, as the green color, indicative of this bacteria colonization following a 24-h incubation, has vanished. A 5-min long treatment leads to the full inactivation of the three bacteria strains in the scanned region, while some circle shaped colonies are observed on the edges of the treated zone. These circle shape colonies are likely to be assigned to *E. coli* persistence, as this specific morphology was already observed in the single bacteria inoculation experiment documented in Figure 10. A 10-min plasma treatment leads to the

full inactivation of the bacterial inoculum with no evidence for any of the three bacterial strains growth. The plasma treatment of the mixed inoculum reveals that this inactivation method is likely to be efficient for the decontamination of multi-infected wounds. Finally, full decontamination was also achieved for S. dysgalactiae and B. subtilis strains collected at Orleans Hospital (data not shown).

*3.5. Pilot Tolerance Study on the Use of Plasma Therapy for Wound Treatment*

This preliminary study was performed at Orleans Regional Hospital (CHRO) in the department of Tropical and Infectious Diseases under the supervision of Dr T. Prazuck. The study was approved by the CHRO ethical committee on the base of bibliographic data published by different teams reporting clinical trials with non-thermal plasma devices and following a tolerance study performed at Institut Gustave Roussy (Villejuif, France) under the supervision of Dr L. Mir [62], with the multi-jet applicator technology described in this work applied to both nude and hairy mice. It was measured that neither severe nor irreversible skin disorders were induced following minutes long multi-jet delivery in a single fraction. The tolerance study performed at CHRO included five hospitalized patients bearing superinfected wounds having diameters ranging from 3 to 15 cm. A single jet low voltage device was used to deliver plasma during one minute for each half square centimeter surface unit, as shown in Figure 12. The full time exposure of the wound treated zone, which was not always covering the whole ulcerated tissue, ranged from 15 to 30 min. The plasma device was handheld and operated either by the coauthors or the medical staff of the hospital. Each of the five patients was plasma treated either daily or each other day, reaching at the end a median five-day fraction of plasma treatment. The patients were preventively grounded by means of an antistatic bracelet so to better control their potential and the plasma treatment. Tolerance was assessed by scoring the patient feelings during and after the plasma delivery fractions. It was also systematically and carefully investigated by nurses in charge of the treatment that no necrotic, inflammatory and cutaneous stress appeared at the place where the plasma jet was applied. As reported in Figure 12, the temperature of the treated area was monitored live by means of an infrared camera (Fluke Ti480 by Fluke). The surface of the wound is in general colder due to the presence of liquid and the consequent cooling caused by evaporation. At the place where the plasma jet was applied a temporary small increase (about +3–5 °C) of the surface temperature was observed. The recorded temperature always remained below 40 °C, so any adverse effect due to excessive heating can be discarded.

The tolerance study was overall scored as excellent. The scoring, associated with any local or systemic pain felt during or after the plasma fraction, remained at the same value as that expressed before plasma treatment for each of the patients or for all of the plasma fractions. This testifies to the absence of pain, burn or any unpleasant feeling by the patient during the plasma delivery on injured skin. No inflammation, necrosis or degradation of the ulcerated tissue was reported.

Figure 13 presents the *Staphylococcus aureus ATCC9144* and *Pseudomonas aeruginosa ATCC9027* colony reduction versus the plasma exposure duration. For the two bacteria, three different initial concentrations were exposed in triplicate: respectively about 170, 90 and 50 and 160, 55, 30 colonies/membrane were used for Staphylococcus aureus and Pseudomonas aeruginosa respectively. These colony concentrations were selected as being representative of typical infection levels in the hospital context following the baseline decontamination protocols using disinfectants. In Figure 13, the colony reduction was plotted after applying a normalization for three experiments (realized in triplicate) for each of the bacterial strains. It is worth noting that whatever the initial colony concentration, the same trends are measured, leading to an efficient reduction after a 10-min-long exposure for both bacterial strains.

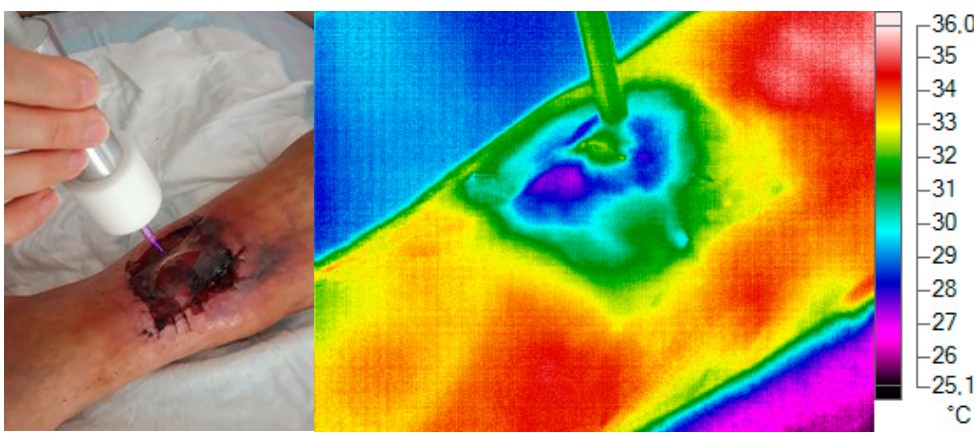

**Figure 12.** Photo of the delivery of a single jet generated with the low voltage applicator on a patient's wound included in the pilot tolerance study performed at CHRO, (right) the treatment monitored by means of an IR camera.

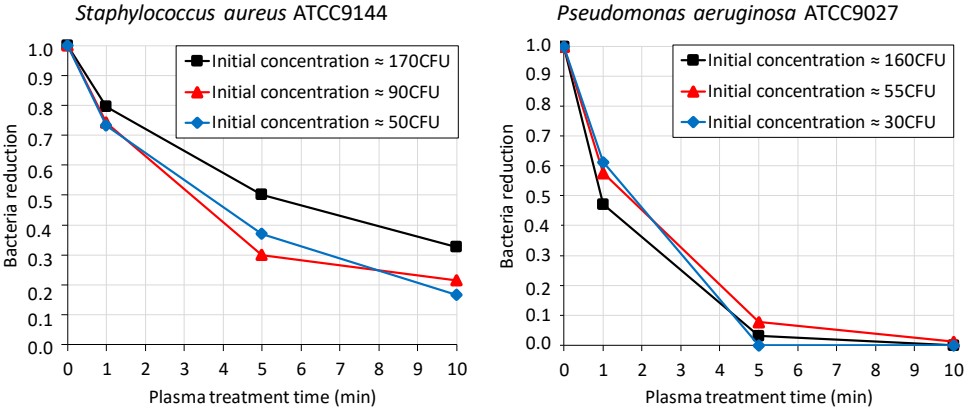

**Figure 13.** Normalized colony reduction versus plasma treatment time with the diffuse mode plasma applicator for (**left**) *Staphylococcus aureus ATCC9144* and (**right**) *Pseudomonas aeruginosa ATCC9027* strains. The experiments were performed in triplicate with three different initial colony concentrations for each strain.

## 4. Discussion

This work reports on the design and operation of multi jets devices based on the plasma gun technology [42,43].

The first device developed as a "multi-spot plasma applicator" and a slightly modified derived version labeled as a "diffuse mode plasma applicator" allows for the delivery of 52 mm-sized helium jets in ambient air at a pulse repetition rate of 2 kHz. In the diffuse mode operation, additional plasma volume is generated in between the 52 multi jets. These two applicators have been developed and used to plasma treat bacteria contaminated substrates either 55 mm in diameter or consisting in large $12 \times 12$ cm$^2$ samples. The main objective was to assess the antibacterial action of such multi jets for various bacteria, including drug resistant strains. Such plasma potentiality was already reported in the literature, but this work addresses simultaneously various plasma delivery strategies for decontamination applications in the perspective of infected and chronic wounds treatment in hospital centers. Indeed, four main plasma exposure protocols have been experienced and demonstrated: plasma treatment of agar samples right after their inoculation with various single bacteria strain, plasma treatment of mature bacterial lawn grown on agar substrates, plasma treatment of multi-contaminated with three bacteria strains samples, and multi jet treatment in static and scanning modes over extended surface samples.

The second class of multi jet device assessed in this work consisted of a bunch of individual millimeter-sized helium jets generated at the lower voltage of 2.5 kV peak amplitude and a higher repetition rate of 5 to 20 kHz. It has been developed to deliver one single jet, and 4, 9 and 13 secondary helium jets. It is reported that for this low voltage applicator, the power and current are quite equally distributed in between each secondary jet (see Figure 6), and that this "low voltage applicator" allows for safe and painless multi jet exposure even on uneven, non-flat surfaces (see Figure 2). A single jet low voltage plasma jet representative of this second class of plasma device has been used in a pilot clinical tolerance study performed at the Orleans Hospital (CHRO) on superinfected wound bearing patients.

The bacterial decontamination study indicates first that the plasma action is localized in the region impacted by the plasma jets, and it does not diffuse strongly in between the secondary jets regions during plasma jet exposure (see Figure 7). This result is a good indication in the perspective of wound treatment, as the plasma treatment can be restricted to the zone where bacterial decontamination is required, while not exposing healthy tissues. A time exposure dependent inhibition surface is measured on contaminated samples exposed to multi jets. As the diameter of the multi jets is almost constant, when emerging from same diameter channels drilled in the plasma applicators, two main conclusions result from this measurement. First, since the plasma light emission volume remains constant throughout any exposure time, it is concluded that the plasma light emission cross section observed on the sample surface is not the best indicator to predict the inhibition zone diameter. It is proposed that both plasma generated reactive species accumulation and the combination of these reactive species with transient electric fields inherent with the multi-jets delivery may play a role in the bacteria decontamination. In addition to the reactive species delivery, the bacteria on the central zone of the plasma jet impact zone may be inactivated during a short exposure as it is exposed to a rather intense electric field, while those on the more outer locations will require the accumulation of a larger number of lower strength electric field pulses and lower doses of reactive species.

Similarly, and with regard to extended wound treatment, the use of the diffuse mode applicator (see Figure 8) or of the rotating (see Figure 7), linear (see Figure 10) and zigzag (see Figure 11) scanning protocols, demonstrates that large surfaces may be rather homogeneously decontaminated, extending the potentialities of plasma technology for these medical applications. Shorter treatment plasma exposure can indeed be developed on the basis of these multi jets scanning applicators. A few square centimeter large contaminated sample was inactivated for a single fraction of only a one-minute-long exposure. In the context of patient wound treatment, such short time exposure is an excellent indication of a highly tolerable therapeutic protocol, and may be envisioned as a daily fractionated treatment.

Bacterial inactivation was demonstrated for *S. aureus*, *P. aeruginosa* and drug resistant *S. aureus*, *P. aeruginosa* and *E. coli* strains collected for patients' wounds at Orleans hospital. Although not documented in this work, full decontamination was also achieved for the *S. dysgalactiae* and *B. subtilis* strains collected at Orleans Hospital. This confirms that plasma action has a broad spectrum target approach likely to be applied for various bacteria families, either gram positive or gram negative, even with the same plasma operation mode. It is also demonstrated that plasma multi jets can be a potent bactericidal agent for native or mature bacterial lawns contaminated with a single but also a mix of various pathogens. Similar efficiency was demonstrated for antibiotic-resistant and non-resistant *P. aeruginosa* (see Figure 9) as previously reported but for *E. coli* bacteria and with dielectric barrier discharge applicator [33]. This strengthens the well-known unique status of plasma technology in comparison with conventional antibiotic drugs, as a broad spectrum and non-systemic, and thus likely a low side-effect, therapeutic option.

Finally, the low voltage plasma applicator was used for the fractionated treatment of five wound-bearing patients with the focus on the scoring of the treatment tolerance and toxicity assessment. The plasma applicator was used for a few tens of minutes' exposures,

by different nurses, and for wounds of various extensions, surface topologies, including necrotic, dry and humid zones. As summarized in Section 3.5, the tolerance scoring was excellent, with absolutely no pain, skin injury, inflammatory feeling or observed effects. While the study was focused on the tolerance and toxicity of plasma treatment, it was nevertheless observed, even with this limited patient cohort, a trend for the improvement of the tissue status on the plasma treated zones in comparison with the non-targeted zones for the same patients. Work is ongoing to assess the tolerance and healing efficiency of plasma multi-jets (either administered daily or each other day) delivered alone and in combination with dedicated dressings. This work is a new contribution in the development of plasma technology therapy for wound treatment, but as expressed in the conclusion of [40], "the possible role of cold atmospheric pressure plasma in clinical multi drug resistant decontamination must be evaluated in clinical trials with repeated plasma treatment embedded in a comprehensive hygienic decontamination concept". This mandatory step is planned in collaboration with CHRO. This work demonstrates that the on-demand design of multi jets applicators can be performed for the development of various plasma applicator sizes and shapes with various numbers and spatial distributions of the multi jets likely covering very extended surfaces.

**Author Contributions:** Conceptualization, T.M., M.D., S.D., I.G., L.H., C.L.H., J.-M.P, T.P., R.B. and E.R.; methodology, T.M., F.B.-M., M.D., S.D., P.E.B., I.G., C.L.H., J.-M.P., A.S. and E.R.; investigation, T.M., M.D., S.D., I.G., C.L.H., I.O., C.T. and E.R.; resources, M.D., C.L.H. and T.P.; writing—original draft preparation, T.M., T.P., J.-M.P. and E.R.; writing—review and editing, all co-authors; supervision, M.D., T.P., J.-M.P. and E.R.; funding acquisition, C.D., J.-M.P. and E.R. All authors have read and agreed to the published version of the manuscript.

**Funding:** This research was funded by PEPS-CNRS project 'ACUMULTIPLAS'.

**Institutional Review Board Statement:** Ethical review and approval were waived for this study, due to focus of the pilot study on tolerance assessment performed with informed consent of patients and with a gradual increase of the plasma exposure time.

**Informed Consent Statement:** Informed consent was obtained from all subjects involved in the study.

**Data Availability Statement:** The data that support the methods and findings of this study are available from the corresponding author upon reasonable request.

**Acknowledgments:** This work was supported by PEPS-CNRS 'ACUMULTIPLAS', CNRS GDR 2025 'HAPPYBIO'. Authors express their sincere acknowledgments to João SANTOS SOUSA (LPGP, Université Paris-Saclay, CNRS, 91405 Orsay, France), Lluis. M. MIR (Institut Gustave Roussy, Metabolic and Systemic Aspects of Oncogenesis (METSY), Université Paris-Saclay, CNRS, 94805 Villejuif, France), Thai Hoa CHUNG (Institut Gustave Roussy, Metabolic and Systemic Aspects of Oncogenesis (METSY), Université Paris-Saclay, CNRS, 94805 Villejuif, France) for their contribution in preliminary but key experiments and discussions during the PEPS 'ACUMULTIPLAS"s project. I.O. is supported by 'PLASFECT' project funded through the 2nd French-German call for projects on antimicrobial resistance 2020 "One Health: AMR in environmental reservoirs and Colonizing antibiotic-resistant bacteria. A.S. is supported by 'MINIONS' project funded through ARD Centre Val De Loire COSMETOSCIENCES.

**Conflicts of Interest:** The authors declare no conflict of interest. The funders had no role in the design of the study; in the collection, analyses, or interpretation of data; in the writing of the manuscript, or in the decision to publish the results.

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
