# Peer review of "Anti-Bacterial Action of Plasma Multi-Jets in the Context of Chronic Wound Healing"

_applsci, doi:10.3390/app11209598_

Round 1

Reviewer 1 Report

The paper is about one of “hot topics” of plasma applications, the field of plasma medicine and is very actual and important. A detailed description of the methods and experiments is presented. A pilot tolerance study is performed to five hospitalized patients which is very important step in plasma medicine field.

In the same time there are some weaknesses that need more detailed explanations and clarifications.

The main problem is that the results are only photos. There are not any other quantitative results presented in addition to the photos (graphs, tables). The only graph presenting the current in Fig. 6a is of very bad quality. Are there any measurements (size of the spots, concentration of the bacteria in different yellow regions)?

There are some more not enough clear things in the paper.

Can you clarify If the part of the plasma that is in contact with the treated samples corresponds to the active plasma region or it is an afterglow region? What is the reason for plasma appearance outside the tubes: because of the gas flow or there is another reason?

The size of the treated surfaces is not presented in most of the experiments. Can you clarify it?

Would you clarify the experimental conditions in Fig. 7d? What is the meaning in the text “…the impact of the 4, 12 and 12 secondary jets, …”? Are the secondary jets acting in the same time or the 4, 12 and 12 are switched on successively, for how long time, ets.?

Are there experiments with the diffuse mode applicator only but without the jets? It would be interesting to see its separated effect. Otherwise, the text on p.12 is nothing but the authors ideas that has not been proved.

The results presented in Fig. 10 – the experiment is not correct because of the possible affect of different areas. It is not possible to understand from the picture the size of the treated area, the way of the multiple treatment.

Row 611 – “exposed to rather intense electric field”. Are there any measurements of the electric field?

The “Discussion” part is actually a summary of the paper but not a discussion. It would be better to transform the “Results” and “Discussion” into one single part “Results and discussion” shortening the paper and avoiding repeating the texts.

And there is not any “Conclusion” part in the paper. Maybe some texts from “Discussion” would be better to appear in “Conclusion”.

Author Response

Dear Editor,

We thank the two referees for their positive comments and give below a point by point answer to their questions. We hope to fit all their requirements and believe our manuscript is now more complete and suitable for publication.

First referee

The only graph presenting the current in Fig. 6a is of very bad quality.

The same graph but with a much higher resolution has been submitted, sorry for the original version of this one.

Are there any measurements: size of the spots,

The sum of the size of the plasma spots have been indicated for figure 7d (as a representative example).

Are there any measurements: concentration of the bacteria in different yellow regions)?

This was unfortunately not performed at the time of the experimental campaign. An additional figure (fig.13) has been added, giving some quantitative measurement of the bacteria colony reduction, in the context of hospital infection.

The following text and figure caption have been introduced at the end of section 3.5.

“The figure 13 presents the Staphylococcus aureus ATCC9144 and Pseudomonas aeruginosa ATCC9027 colony reduction versus the plasma exposure duration. For the two bacteria, three different initial concentrations were exposed in triplicate: respectively about 170, 90 and 50 and 160, 55, 30 colonies/membrane were used for Staphylococcus aureus and Pseudomonas aeruginosa respectively. These colony concentrations were selected as being representative or typical infection level in the hospital context following the baseline decontamination protocols using disinfectant. In figure13, the colony reduction was plotted after applying a normalization of for three experiments for each of the bacterial strain. One note that whatever the initial colony concentration, the same trends are measured, leading to an efficient reduction after a 10-minute-long exposure for both bacterial strains.”

Figure 13 Normalized colony reduction versus plasma treatment time with the diffuse mode plasma applicator for a) Staphylococcus aureus ATCC9144 and b) Pseudomonas aeruginosa ATCC9027 strains. The experiments were performed in triplicate with three different initial colony concentrations for each strain.

Can you clarify If the part of the plasma that is in contact with the treated samples corresponds to the active plasma region or it is an afterglow region?

The following sentence has been added in section 2.1, right above the figure 1, to clarify that the samples are exposed to the “plasma plume” and not an afterglow. This is confirmed by current measurements, as documented in Fig 6a. One observe that the current pulse is composed of a very short spike, associated with the contact of the plasma with the target, and a much longer duration current contribution, connected with the voltage pulse waveform (2 µs FWHM in this work).

“The plasma in contact with the sample is associated with ionization wave propagation in the helium gas flow at the outlet of the capillary.”

What is the reason for plasma appearance outside the tubes: because of the gas flow or there is another reason?

As mentioned before, the plasma outside the tubes is the so-called “plasma plume” expanding in the helium rich region at the capillary exit.

The size of the treated surfaces is not presented in most of the experiments. Can you clarify it?

The treated surface for Fig.7, 8 and 9 was of 10 cm2. This was added in section 2.21, line 210 as:

“Following a 24-hour incubation period at 37°C, a 10 cm2 bacterial lawn was produced as shown in figure 3d.”

We also add the following information about figure 7d (chosen as an example for plasma treated surface assessment).

“In figure 7d, it is observed that the impact of the respectively 4, 12 and 12 secondary jets -distributed from the center to the periphery of the plasma applicator- results in a full inactivation of the bacteria. Because of an imperfect centering of the plasma applicator with respect to the sample, only 4 secondary jets distributed along the 38 mm in diameter outer ring of the multi spot applicator impacted the bacterial lawn, which was about 40 mm in diameter. This is visible in the top right region of figure 7d. The ratio of the sum of these 4, 12, 12 and 4 plasma spot impact surfaces over the bacterial lawn surface results in about an 8% coverage for a 5-minute-long static exposure with the multi spot applicator. Thus, the decontamination of the full surface for the extended bacterial lawn would require a displacement of the plasma applicator, as will be documented in section 3.4”

The treated surface for the 12x12 cm2 bacteria lawn was of about 40 cm2 with the translation scanning mode and of about 32 cm2 for the zigzag scanning protocol. This has been added in Fig.10 and Fig.11 captions.

Would you clarify the experimental conditions in Fig. 7d?

In Fig 7d) caption we indicated “H24 image of the sample exposed at H0 to 5 min plasma treatment in static mode”

This means that the multi spot plasma applicator delivered all the secondary jets at H0, defined as the time origin, and that images was done 24 hours later (H24).

What is the meaning in the text “…the impact of the 4, 12 and 12 secondary jets, …”? Are the secondary jets acting in the same time or the 4, 12 and 12 are switched on successively, for how long time, ets.?

All the secondary jets are delivered with the multi spot plasma applicator at the same time. To clarify this point, the following sentence has been added in 2.1, right above the figure 1:

“With the multi spot plasma applicator, all the secondary jets distributed along the 8,18, 28 and 38 mm in diameter rings were delivered simultaneously”

Are there experiments with the diffuse mode applicator only but without the jets? It would be interesting to see its separated effect. Otherwise, the text on p.12 is nothing but the authors ideas that has not been proved.

No, unfortunately, it is not possible to use the diffuse plasma only. As we mentioned, the secondary jets are always active but with the additional sleeve, a diffuse plasma is generated together with these secondary jets.

In Page 12 we wrote: “Third, the yellow color, representative for the S. aureus colonization, is less intense in between the multi-jet impact regions in figure 8 than it was in figure 7. This indicates that the growth of bacterial lawn even in the region in between the multi spots is reduced with the diffuse mode applicator, presumably because of reactive species’ diffusion from impact plasma points thanks to additional sleeve. These two latter observations reveal that the diffuse mode applicator modify the relative bactericidal efficiency of the individual multi-jets and that the whole surface of the sample is plasma-treated and partly inactivated even though the multi-jets are much more effective than the diffuse plasma”

This is not any speculation, but observation that the bacterial lawn is less colored in between the secondary jet impact spots in comparison with images in figure 7.

And we wrote: “The lower efficiency in the central region of the diffuse mode applicator may be because this zone is helium-rich thus less favorable for the generation of reactive species.”

This is more speculative and was suppressed.

The results presented in Fig. 10 – the experiment is not correct because of the possible affect of different areas. It is not possible to understand from the picture the size of the treated area, the way of the multiple treatment.

The size of the treated area (40 cm2) was added, as mentioned before. We agree that on the boarders of the 1, 5 and 10-minute exposure zones, there may be some overlap of the treatment with the neighbored zones. But as seen in figure 11, that depicts results for a single treatment, the decontamination surface clearly corresponds to the treated area. This was added in the manuscript together with the indication that due to this likely overlap, the results are discussed for the central band of each of the three different treatment duration zones.

We add in section 3.4, right below the figure 10:

“There may be some overlap of the treatment in one zone (for 1, 5 and 10 minutes’ plasma exposure) with the neighbored zones. Because of this possible artifact, the decontamination action is discussed for the central band of each of the three different treatment duration zones.”

Row 611 – “exposed to rather intense electric field”. Are there any measurements of the electric field?

The electric field amplitude measurement or modeling for plasma jets were published in references [45, 53-57] as mentioned in the manuscript. With our experimental set-up, electric field with amplitudes ranging from 3 to 8 kV/cm were reported along the plasma jet inside the capillary [45], while amplitude as high as 45 kV/cm were measured at the tip of the plasma plume [55].

For reader’s information we add

“With our experimental set-up, and with a slightly higher peak voltage amplitude applied to the plasma reactor 16 kV instead of 14 kV in this work, electric field with amplitudes ranging from 3 to 8 kV/cm were reported along the plasma jet inside the capillary [45], while amplitude as high as 45 kV/cm were measured at the tip of the plasma plume [55].”

The “Discussion” part is actually a summary of the paper but not a discussion. It would be better to transform the “Results” and “Discussion” into one single part “Results and discussion” shortening the paper and avoiding repeating the texts.

And there is not any “Conclusion” part in the paper. Maybe some texts from “Discussion” would be better to appear in “Conclusion”.

We follow the template of the journal which indicate that a conclusion section is only required if the mandatory discussion section is excessively long. In our manuscript, the discussion section aims to summarize the main results of our study, and this section is not long enough to require a summary as a conclusion.

Second referee

“…Along these lines, I would ask that the authors include an estimated coverage area for the jets. This would be a ratio of the estimated treated area/the size of the plasma applicator”

We make this assessment for figure 7d and add the following information in the manuscript:

“The ratio of the sum of these 4, 12, 12 and 4 plasma spot impact surfaces over the bacterial lawn surface results in about an 8% coverage for a 5-minute-long static exposure with the multi spot applicator. Thus, the decontamination of the full surface for the extended bacterial lawn would require a displacement of the plasma applicator, as will be documented in section 3.4”

Minor comments:

  1. Line 350 says impact of 4, 12 and 12 secondary jets. Is this supposed to be and 24 secondary jets? it is confusing to see which 12 are being discussed.

We wrote: “In figure 7d, it is observed that the impact of the respectively 4, 12 and 12 secondary jets -distributed from the center to the periphery of the plasma applicator- results in a full inactivation of the bacteria”

This is indeed the plasma spot impact of the 4,12 and 12 secondary jets distributed along the 8,18 and 28 mm in diameter rings of the multi spot plasma applicator. The outer 24 secondary jets distributed along the 38 mm ring are almost all outside the bacterial lawn which was generated over the 50 in mm diameter membrane. Only 4 of these outer secondary jets exposed the bacterial lawn, as seen on the top right region in figure 6d.

To clarify this, we change the text as follow:

“In figure 7d, it is observed that the impact of the respectively 4, 12 and 12 secondary jets -distributed from the center to the periphery of the plasma applicator- results in a full inactivation of the bacteria. Because of an imperfect centering of the plasma applicator with respect to the sample, only 4 secondary jets distributed along the 38 mm in diameter outer ring of the multi spot applicator impacted the bacterial lawn, which was about 40 mm in diameter. This is visible in the top right region of figure 7d”.

Reviewer 2 Report

Dear authors,

The manuscript does a great job of introducing the three different jet devices and detailing their differences between each other. The antimicrobial activity assessed between the jets shows their overall area of removal from a bacterial lawn between resistant and sensitive cultures and does a good job of explaining the differences observed. I believe this is good for publication for Applied Sciences as an intro to the various ways jet plasma can be applied for wound healing to cover the entire wound surface. Along these lines, I would ask that the authors include an estimated coverage area for the jets. This would be a ratio of the estimated treated area/the size of the plasma applicator. I realize this is difficult to measure due to the observation of complete bacterial removal, partial bacterial removal, and no removal at all, but I think it would help summarize the efficacy differences of the jets. There is enough language in the paper already that suggests this as the plasma applicator area, plasma treated area, and untreated area measurements are listed throughout the paper.

Minor comments:

1. Line 350 says impact of 4, 12 and 12 secondary jets. Is this supposed to be and 24 secondary jets? it is confusing to see which 12 are being discussed.

Author Response

Dear Editor,

We thank the two referees for their positive comments and give below a point by point answer to their questions. We hope to fit all their requirements and believe our manuscript is now more complete and suitable for publication.

First referee

The only graph presenting the current in Fig. 6a is of very bad quality.

The same graph but with a much higher resolution has been submitted, sorry for the original version of this one.

Are there any measurements: size of the spots,

The sum of the size of the plasma spots have been indicated for figure 7d (as a representative example).

Are there any measurements: concentration of the bacteria in different yellow regions)?

This was unfortunately not performed at the time of the experimental campaign. An additional figure (fig.13) has been added, giving some quantitative measurement of the bacteria colony reduction, in the context of hospital infection.

The following text and figure have been introduced at the end of section 3.5.

“The figure 13 presents the Staphylococcus aureus ATCC9144 and Pseudomonas aeruginosa ATCC9027 colony reduction versus the plasma exposure duration. For the two bacteria, three different initial concentrations were exposed in triplicate: respectively about 170, 90 and 50 and 160, 55, 30 colonies/membrane were used for Staphylococcus aureus and Pseudomonas aeruginosa respectively. These colony concentrations were selected as being representative or typical infection level in the hospital context following the baseline decontamination protocols using disinfectant. In figure13, the colony reduction was plotted after applying a normalization of for three experiments for each of the bacterial strain. One note that whatever the initial colony concentration, the same trends are measured, leading to an efficient reduction after a 10-minute-long exposure for both bacterial strains.”

Figure 13 Normalized colony reduction versus plasma treatment time with the diffuse mode plasma applicator for a) Staphylococcus aureus ATCC9144 and b) Pseudomonas aeruginosa ATCC9027 strains. The experiments were performed in triplicate with three different initial colony concentrations for each strain.

Can you clarify If the part of the plasma that is in contact with the treated samples corresponds to the active plasma region or it is an afterglow region?

The following sentence has been added in section 2.1, right above the figure 1, to clarify that the samples are exposed to the “plasma plume” and not an afterglow. This is confirmed by current measurements, as documented in Fig 6a. One observe that the current pulse is composed of a very short spike, associated with the contact of the plasma with the target, and a much longer duration current contribution, connected with the voltage pulse waveform (2 µs FWHM in this work).

“The plasma in contact with the sample is associated with ionization wave propagation in the helium gas flow at the outlet of the capillary.”

What is the reason for plasma appearance outside the tubes: because of the gas flow or there is another reason?

As mentioned before, the plasma outside the tubes is the so-called “plasma plume” expanding in the helium rich region at the capillary exit.

The size of the treated surfaces is not presented in most of the experiments. Can you clarify it?

The treated surface for Fig.7, 8 and 9 was of 10 cm2. This was added in section 2.21, line 210 as:

“Following a 24-hour incubation period at 37°C, a 10 cm2 bacterial lawn was produced as shown in figure 3d.”

We also add the following information about figure 7d (chosen as an example for plasma treated surface assessment).

“In figure 7d, it is observed that the impact of the respectively 4, 12 and 12 secondary jets -distributed from the center to the periphery of the plasma applicator- results in a full inactivation of the bacteria. Because of an imperfect centering of the plasma applicator with respect to the sample, only 4 secondary jets distributed along the 38 mm in diameter outer ring of the multi spot applicator impacted the bacterial lawn, which was about 40 mm in diameter. This is visible in the top right region of figure 7d. The ratio of the sum of these 4, 12, 12 and 4 plasma spot impact surfaces over the bacterial lawn surface results in about an 8% coverage for a 5-minute-long static exposure with the multi spot applicator. Thus, the decontamination of the full surface for the extended bacterial lawn would require a displacement of the plasma applicator, as will be documented in section 3.4”

The treated surface for the 12x12 cm2 bacteria lawn was of about 40 cm2 with the translation scanning mode and of about 32 cm2 for the zigzag scanning protocol. This has been added in Fig.10 and Fig.11 captions.

Would you clarify the experimental conditions in Fig. 7d?

In Fig 7d) caption we indicated “H24 image of the sample exposed at H0 to 5 min plasma treatment in static mode”

This means that the multi spot plasma applicator delivered all the secondary jets at H0, defined as the time origin, and that images was done 24 hours later (H24).

What is the meaning in the text “…the impact of the 4, 12 and 12 secondary jets, …”? Are the secondary jets acting in the same time or the 4, 12 and 12 are switched on successively, for how long time, ets.?

All the secondary jets are delivered with the multi spot plasma applicator at the same time. To clarify this point, the following sentence has been added in 2.1, right above the figure 1:

“With the multi spot plasma applicator, all the secondary jets distributed along the 8,18, 28 and 38 mm in diameter rings were delivered simultaneously”

Are there experiments with the diffuse mode applicator only but without the jets? It would be interesting to see its separated effect. Otherwise, the text on p.12 is nothing but the authors ideas that has not been proved.

No, unfortunately, it is not possible to use the diffuse plasma only. As we mentioned, the secondary jets are always active but with the additional sleeve, a diffuse plasma is generated together with these secondary jets.

In Page 12 we wrote: “Third, the yellow color, representative for the S. aureus colonization, is less intense in between the multi-jet impact regions in figure 8 than it was in figure 7. This indicates that the growth of bacterial lawn even in the region in between the multi spots is reduced with the diffuse mode applicator, presumably because of reactive species’ diffusion from impact plasma points thanks to additional sleeve. These two latter observations reveal that the diffuse mode applicator modify the relative bactericidal efficiency of the individual multi-jets and that the whole surface of the sample is plasma-treated and partly inactivated even though the multi-jets are much more effective than the diffuse plasma”

This is not any speculation, but observation that the bacterial lawn is less colored in between the secondary jet impact spots in comparison with images in figure 7.

And we wrote: “The lower efficiency in the central region of the diffuse mode applicator may be because this zone is helium-rich thus less favorable for the generation of reactive species.”

This is more speculative and was suppressed.

The results presented in Fig. 10 – the experiment is not correct because of the possible affect of different areas. It is not possible to understand from the picture the size of the treated area, the way of the multiple treatment.

The size of the treated area (40 cm2) was added, as mentioned before. We agree that on the boarders of the 1, 5 and 10-minute exposure zones, there may be some overlap of the treatment with the neighbored zones. But as seen in figure 11, that depicts results for a single treatment, the decontamination surface clearly corresponds to the treated area. This was added in the manuscript together with the indication that due to this likely overlap, the results are discussed for the central band of each of the three different treatment duration zones.

We add in section 3.4, right below the figure 10:

“There may be some overlap of the treatment in one zone (for 1, 5 and 10 minutes’ plasma exposure) with the neighbored zones. Because of this possible artifact, the decontamination action is discussed for the central band of each of the three different treatment duration zones.”

Row 611 – “exposed to rather intense electric field”. Are there any measurements of the electric field?

The electric field amplitude measurement or modeling for plasma jets were published in references [45, 53-57] as mentioned in the manuscript. With our experimental set-up, electric field with amplitudes ranging from 3 to 8 kV/cm were reported along the plasma jet inside the capillary [45], while amplitude as high as 45 kV/cm were measured at the tip of the plasma plume [55].

For reader’s information we add

“With our experimental set-up, and with a slightly higher peak voltage amplitude applied to the plasma reactor 16 kV instead of 14 kV in this work, electric field with amplitudes ranging from 3 to 8 kV/cm were reported along the plasma jet inside the capillary [45], while amplitude as high as 45 kV/cm were measured at the tip of the plasma plume [55].”

The “Discussion” part is actually a summary of the paper but not a discussion. It would be better to transform the “Results” and “Discussion” into one single part “Results and discussion” shortening the paper and avoiding repeating the texts.

And there is not any “Conclusion” part in the paper. Maybe some texts from “Discussion” would be better to appear in “Conclusion”.

We follow the template of the journal which indicate that a conclusion section is only required if the mandatory discussion section is excessively long. In our manuscript, the discussion section aims to summarize the main results of our study, and this section is not long enough to require a summary as a conclusion.

Second referee

“…Along these lines, I would ask that the authors include an estimated coverage area for the jets. This would be a ratio of the estimated treated area/the size of the plasma applicator”

We make this assessment for figure 7d and add the following information in the manuscript:

“The ratio of the sum of these 4, 12, 12 and 4 plasma spot impact surfaces over the bacterial lawn surface results in about an 8% coverage for a 5-minute-long static exposure with the multi spot applicator. Thus, the decontamination of the full surface for the extended bacterial lawn would require a displacement of the plasma applicator, as will be documented in section 3.4”

Minor comments:

  1. Line 350 says impact of 4, 12 and 12 secondary jets. Is this supposed to be and 24 secondary jets? it is confusing to see which 12 are being discussed.

We wrote: “In figure 7d, it is observed that the impact of the respectively 4, 12 and 12 secondary jets -distributed from the center to the periphery of the plasma applicator- results in a full inactivation of the bacteria”

This is indeed the plasma spot impact of the 4,12 and 12 secondary jets distributed along the 8,18 and 28 mm in diameter rings of the multi spot plasma applicator. The outer 24 secondary jets distributed along the 38 mm ring are almost all outside the bacterial lawn which was generated over the 50 in mm diameter membrane. Only 4 of these outer secondary jets exposed the bacterial lawn, as seen on the top right region in figure 6d.

To clarify this, we change the text as follow:

“In figure 7d, it is observed that the impact of the respectively 4, 12 and 12 secondary jets -distributed from the center to the periphery of the plasma applicator- results in a full inactivation of the bacteria. Because of an imperfect centering of the plasma applicator with respect to the sample, only 4 secondary jets distributed along the 38 mm in diameter outer ring of the multi spot applicator impacted the bacterial lawn, which was about 40 mm in diameter. This is visible in the top right region of figure 7d”.
